# What influences uptake and early adherence to Option B+ (lifelong antiretroviral therapy among HIV positive pregnant and breastfeeding women) in Central Uganda? A mixed methods study

**Aggrey David Mukose**[1,2]*, **Hilde Bastiaens**[2,3], **Fredrick Makumbi**[1], **Esther Buregyeya**[4], **Rose Naigino**[5], **Joshua Musinguzi**[6], **Jean-Pierre Van Geertruyden**[2], **Rhoda K. Wanyenze**[4]

1 Department of Epidemiology and Biostatistics, School of Public Health, College of Health Sciences, Makerere University, Kampala, Uganda, 2 Global Health Institute, Department of Epidemiology and Social Medicine, University of Antwerp, Antwerp, Belgium, 3 Department of Primary and Interdisciplinary Care, University of Antwerp, Antwerp, Belgium, 4 Department of Disease Control and Environmental Health, School of Public Health, College of Health Sciences, Makerere University, Kampala, Uganda, 5 Makerere University School of Public Health, Kampala, Uganda, 6 Ministry of Health, Kampala, Uganda

* amukose@musph.ac.ug

**Data Availability Statement:** Due to restrictions by the Makerere University School of Public Health

## Abstract

### Background

High uptake and optimal adherence to Option B+ antiretroviral therapy (ART) increase effectiveness in averting mother-to-child transmission of HIV. Option B+ ART uptake, early adherence, and associated factors need to be evaluated in Central Uganda.

### Methods

A mixed approaches study was carried out in six health facilities in Masaka, Mityana, and Luwero districts from October 2013 to February 2016. Questionnaires were administered to 507 HIV positive pregnant females seeking antenatal care services. Key informant interviews were conducted with 54 health providers, and in-depth interviews (IDIs) with 57 HIV positive women on Option B+ ART. Quantitative data were analyzed using log-binomial regression model to determine factors associated with optimal adherence (taking at least 95% of the prescribed ART), while thematic analysis was used on qualitative data.

### Results

Ninety one percent of women (463/507) received a prescription of life long ART. Of these, 93.3% (432/463) started swallowing their medicines. Overall, 83% of women who received ART prescriptions (310/374) felt they were ready to initiate ART immediately. Main motivating factors to swallow ART among those who received a prescription were women's personal desire to be healthy (92.3%) and desire to protect their babies (90.6%). Optimal adherence to ART was achieved by 76.8% (315/410). Adherence was higher among

Higher Degrees Research and Ethics Committee, some access restrictions apply to the data for reasons of safety and protection of study subjects and their institutions. Sensitive data was collected from patients and they didn't consent to open data access. However, criteria eligible researchers with interest in the data may request for anonymized data access through the Chair Higher Degrees, Research and Ethics Committee. The contact information for the ethics committee to which data requests may be sent is: The Chairperson Makerere University School of Public Health Higher Degrees, Research and Ethics Committee, P.O. Box 7072, Kampala. Telephone +256414 532207/543872/543437.

**Funding:** This study was funded by the Global Fund through the Ministry of Health-Uganda [Grant Number: UGD-708-G07-H]. The funders had no role in study design, data collection and analysis, decision to publish, or preparation of the manuscript.

**Competing interests:** The authors have declared that no competing interests exist.

females who were ready to start ART (adj. PR = 3.20; 95% CI: 1.15–8.79) and those who had revealed their HIV positive result to someone (adj. PR = 1.23; 95% CI: 1.04–1.46). Facilitators of ART uptake from qualitative findings included adequate counseling, willingness to start, and knowing the benefits of ART. Reasons for refusal to start ART included being unready to start ART, fear to take ART for life, doubt of HIV positive results, and preference for local herbs. Reasons for non-adherence were travelling far away from health facilities, fear of side effects, non-disclosure of HIV results to anyone, and perception that the baby is safe from HIV infection post-delivery.

## Conclusions

Uptake of Option B+ ART was very high. However, failure to start swallowing ART and suboptimal adherence are a major public health concern. Enhancing women's readiness to start ART and encouraging HIV result revelation could improve ART uptake and adherence.

## Introduction

Mother-to-child transmission (MTCT) of HIV is still a challenge sub-Saharan Africa where HIV prevalence and fertility rates are high [1, 2]. Globally, over 90% of new HIV infections in children occur through MTCT [3]. ART use for prevention of mother-to-child transmission (PMTCT) lessens the HIV MTCT risk to less than 2% and 5% in non- and lactating populations respectively [4, 5]. Since 2000, a decline of 58% in new paediatric HIV infections has been noted as a result of ART use among expectant women living with HIV [1, 6, 7]. To hasten the drop in new HIV infections more, the World Health Organization (WHO) reviewed the PMTCT guidelines in 2012 with the aim of achieving virtual elimination of MTCT (e-MTCT) of HIV. The reviewed guidelines suggested Option B+ (lifelong ART for HIV positive expecting and lactating mothers irrespective of CD4 or clinical stage) [8].

Implementation of Option B+ in Uganda began in September 2012. Implementation started in districts with the highest HIV prevalence and was scaled up to all districts by the end of 2013 [9]. Women on Option B+ in Uganda received a combination of Tenofovir Disoproxil Fumarate (TDF), Lamivudine (3TC) and Efavirenz (EFV) as a single pill. For e-MTCT of HIV to be realized, ART should be initiated early in pregnancy with adequate adherence in order to achieve viral suppression [10]. However, studies have revealed various gaps in implementation of PMTCT programs. A systematic review and meta-analysis by Huang et al. [11] showed that in China, 71% of HIV positive women had initiated Option B+ but uptake varied between income levels. Further, a systematic assessment in low and middle-income countries revealed that nearly half of HIV positive expectant women neither received ART prophylaxis during antenatal care (ANC) nor delivered in health facilities [12]. Other PMTCT studies conducted before adoption of Option B+ revealed varying results. In Malawi, there was low use of ART and single dose Nevirapine, whereas in Kenya high uptake of ART was reported using community-based assessment data [13, 14]. A study using health facility data from 2002 to 2011 in Northern Uganda showed that only 69.4% and 9.6% of HIV positive gravid women were started on ART for prophylaxis and treatment respectively [15].

Price et al. (2014) specifically evaluated the Option B+ in Malawi using a retrospective cohort where many eligible pregnant women had not taken up an invite to start ART [16]. Gravidity and the first months of postpartum are associated with significant changes in a

woman's life, which might influence ART uptake and adherence. Barriers to ART uptake and adherence during this period among others include family obligations and stress due to pregnancy and child birth, and non-disclosure of HIV positive status. Postpartum women have been found to miss more medical appointments, and adherence tends to be lower than during pregnancy [17–20]. Studies have shown that between 10% and 50% stopped their ART after delivery, some without health workers' approval [18, 19, 21]. Although critical, issues around prescription, uptake of ART (prescription and starting to swallow ART), and early adherence in the setting of Option B + have scarcely been researched [22, 23]. Though many of the existing studies used prescription and receipt of ART as a measure of uptake, some women who receive the medicines could fail to swallow them for various reasons. If ART non- uptake and non-adherence are identified early, interventions can be instituted to increase uptake, adherence and subsequently effectiveness of Option B+. Early treatment failure and hasty progress to ART drug resistance could also be prevented [24, 25]. Therefore, this study assessed key issues around ART prescription and swallowing (uptake), early adherence and associated factors among HIV positive expectant women and lactating mothers on Option B+ in Central Uganda. Findings from the study could contribute to achievement of the Joint United Nations Programme on HIV/AIDS (UNAIDS) fast-track targets and subsequently end the HIV/AIDS epidemic as a public health threat by 2030 [26, 27].

## Methods

### Study design

A prospective cohort study using mixed methods was conducted at health facilities providing lifelong ART in Masaka, Mityana, and Luwero districts from October 2013 to March 2016. The details of the methodology for the study involving HIV positive women are already reported elsewhere [28, 29]. In brief, quantitative interviews were conducted in a prospective cohort of 507 HIV positive women pregnant at baseline and at 2, 4, 6, 10, 14, and 18 months after baseline to establish uptake of ART and other PMTCT linked services for both the HIV positive mothers and their infants; evaluate retention in care; and measure adherence to ART. Additionally, the study enrolled 54 health providers and 57 HIV positive women for a qualitative inquiry using a qualitative descriptive approach. In the current manuscript, a mixed methods cross-sectional convergent parallel study design nested in a prospective cohort was followed. Mixed methods were critical in this study to ensure complementarity and facilitation of greater understanding [30–32] of uptake and early adherence to Option B+ ART. Convergent parallel mixed method design enabled autonomous collection of quantitative and qualitative data which were later integrated at reporting, interpretation, and discussion [30, 33–35]. The quantitative arm aimed to assess issues around ART prescription, uptake, adherence and associated factors. The qualitative arm explored experiences around ART prescription, uptake, and adherence among women on Option B+ and garnered views and perspectives of Option B + service providers.

### Study sites

This study was conducted at six health facilities in Masaka, Mityana, and Luwero districts in Central Uganda. These were among the first districts to implement Option B+ in Uganda following its launch in September 2012. The facilities included three high patient volume facilities (Masaka Regional Referral Hospital (RRH), Mityana General Hospital (GH), and Luwero Health Centre (HC) IV) in accordance with the structure of the Uganda public health-care system [36]. The other three locations were low patient volume facilities (Kyanamukaka HC IV, Ssunga HC III and Katikamu HC III). High patient volume facilities have over 500 individuals

in HIV care while low volume facilities have ≤500 clients in HIV care [37]. The qualitative study was conducted at all the six facilities whereas the quantitative component was carried out only at the high patient volume facilities.

## Study population and data collection

**Quantitative component.** A total of 507 HIV positive pregnant females were enrolled and followed-up at Masaka RRH (184), Mityana GH (213), and Luwero HC IV (110) from October 2013 to March 2016. Study eligibility was HIV positive expectant females seeking ANC services, aged 15–49 years, ART naïve or started on ART for Option B+ within four weeks of enrollment. All women reporting for ANC services at the facilities were assessed for pregnancy either through self-report or pregnancy test and screened for HIV status. The ANC service providers referred all pregnant HIV positive women to the study research staff on site.

A well-trained female research assistant (RA) was stationed at each study facility to enroll and follow up eligible participants. The RA screened the women for study eligibility, administered written informed consent to the eligible women, and subsequently conducted face-to-face interviews using structured pre-tested and standardized questionnaires. The questionnaire used for data collection was adopted with modifications [38–41]. Pre-testing of the questionnaires was done by administering them to a sample of study eligible participants in a non-study site. Thereafter, the investigators met with the RAs to go through the entire tool to correct any unclear questions. This ensured that all participants were asked the same questions in an identical format and responses recorded in a uniform manner. Enrolment was done concurrently at all the three sites until the calculated sample size of 500 participants was obtained. This was the sample size for the parent study and was determined as described elsewhere [29]. Due to simultaneous recruitment at all the study sites, a total of 507 were enrolled. Following a baseline interview at enrolment, follow-up visits were scheduled at every two months intervals up to six months, then every four months up to 18 months post enrolment. Study follow-up interviews were aligned to the scheduled ANC/postnatal care (PNC) clinic visits as much as possible to minimize inconvenience to the women. Baseline questionnaires were administered to all 507 women. Analysis on the characteristics of study participants and prescription of ART was done on 507 women. In this article, uptake of and early adherence to Option B+ medicines at two-month follow-up was analysed for 463 and 410 women who had complete data, respectively. Basing on the sample sizes that we used for the two study outcomes (levels of uptake and early adherence), the study had a 5% level of precision. We conducted a power analysis on associated factors such as HIV positive status disclosure to anyone, and readiness to start ART. We found that our study had sufficient power (≥ 80%) to detect a difference in the study outcome (Early adherence).

Quantitative data collection for this study took place between October 2013 and October 2014.

**Qualitative component.** Participants in the qualitative interviews were purposively selected. We planned to interview 10 HIV positive women at each health facility (60 women total) for the in-depth interviews. However, 57 interviews were conducted and the participants were not part of the quantitative component but receiving the same Option B+ services. Use of in-depth interviews enabled the researchers explore women's lived experience in regards to uptake of and early adherence to ART for Option B+. The women were either expectant or post-delivery. They had been on lifelong ART for a minimum of six months so as to provide sufficient experiences before and after delivery. Based on information from health facility staff and records, the women were considered into three groups: good ART adherers, poor ART adherers, and delayed ART acceptors as defined in the articles that were published elsewhere

[28, 29]. Qualitative data collection was conducted from February to May 2014 using pre-tested semi-structured interview guides. Three well-skilled research assistants with vast under-standing in qualitative research conducted the in-depth interviews.

Fifty-four key informant interviews out of the targeted 60 interviews (10 at each facility) were used to explore health provider perspectives on uptake and early adherence to Option B + from all the six facilities. The selected health providers were deemed to have the necessary information on uptake and early uptake of Option B+ ART. Health providers were classified in two categories based on their roles, workstation/department, and experience as described elsewhere [36]. The sample size specification and justification for the qualitative component were predetermined as this was required by the funder, ethics committee, and Ministry of Health (MOH) before the study was implemented [42, 43].

Four of the main investigators (A.M., R.W., E.B., and R.N.) interviewed the key informants. All in-depth interviews s and a few key informant interviews with expert clients were con-ducted in *Luganda* the commonly spoken local language. All other key informant interviews were carried out in English. On average, each key informant interview lasted 1.5 hours whereas each in-depth interview lasted between 1 and 2 hours. All qualitative interviews were audio-recorded.

## Data collection tools and measures

The quantitative data were collected on participants' socio-demographic characteristics, pre-scription and swallowing of Option B+ ART, reasons for non-prescription and/or refusal to swallow ART, disclosure of HIV status, willingness to start ART, readiness to start taking (ini-tiate) ART, motivation to initiate ART, reasons for delay to initiate ART, understanding of how long the participant was supposed to be on ART, and self-reported adherence to ART. Uptake of and early adherence to Option B+ ART were the key dependent variables. These were assessed at two months after enrolment into the study. *Uptake of Option B+ medicines* was defined as having been prescribed and starting to swallow ART. *Willingness to start ARTs* was assessed among women who had not received a prescription by asking if they were willing to be initiated on ART or not. Reasons for non-willingness to be prescribed ART were also probed. *Readiness to take ART* was assessed by asking women who had been initiated on ART; how ready they were at the time they started taking ART. The responses were; immediately, later and not at all. Women were also asked to mention what motivated them to start taking ART. *Early adherence* was measured as a binary outcome and defined as i) optimal if women reported taking at least 95% of the ART doses (≥29 doses) in the 30-days before the two-month follow-up interview, or ii) suboptimal if reported taking less than 29 doses in the same period. ART was prescribed to be taken once a day. *ART adherence* was assessed through self-report by asking for the number of ART doses taken in the past 30 days. A visual analogue scale (0–100%) was used to find the percentage of prescribed ART doses taken at every follow-up study visit. The questionnaires were translated into Luganda, the commonly used local dialect.

Qualitatively, in-depth interview and key informant interview guides were developed by the research team. The topics covered by the interview guides were based on our research question "What influences uptake and early adherence to lifelong ART among HIV positive pregnant and breastfeeding women in Central Uganda?" Guided by literature gaps, the researchers dis-cussed the issues, listed varied questions and probes in a semi-structured format [28, 36]. This study focused on issues around uptake of, motivation to start and adherence to Option B + ART. Some of the specific questions to the key informants explored matters around lifelong

ART prescription, benefits of lifelong ART, adherence to ART, and retention in HIV care. The interview guide is included as S1 Appendix.

The questions to the in-depth interview participants included; experiences round decision and motivation to start Option B+ ART, refusal to start ART, ART adherence and HIV related stigma. The in-depth interview guide is included as S2 Appendix.

## Data analysis

**Quantitative data.**  Exploratory data analysis was conducted on the independent and dependent variables. Descriptive statistics were generated providing percentages for categorical variables, while means (standard deviation) and median (inter-quartile range, IQR) for continuous variables. Percentages were computed for the socio-demographic characteristics, reported Option B+ ART prescription, taking of ART, reasons for non-prescription and/or refusal to take ART, disclosure of HIV status, willingness to take ART, readiness to initiate ART, motivation for initiating ART, reasons for delay in initiating ART, understanding of how long the participant was supposed to be on ART, and adherence to ART. Bivariate and multivariable analyses were conducted to generate prevalence ratio (PR) as a measure of association, using log-binomial regression model to establish factors associated with optimal adherence. Log-binomial regression was preferred to logistic regression because the later technique tends to overestimate the measures of association when the prevalence of the outcome is 10% or higher [44]. We explored for interaction between some of the factors such as: disclosure of HIV status to anyone; readiness to start ART; health facility level; and knowledge on how long ART should be taken and the outcome (early adherence). After testing for collinearity, interaction and confounding, variables that had p-value of $\leq 0.2$ at bivariate analysis were included into the final multivariable regression model. STATA version 14 was used for analysis.

**Qualitative data.**  Qualitative data were transcribed verbatim. Data in Luganda were simultaneously translated and transliterated into English. Each transcript was reread by AM and one other co-investigator for content and completeness. Additional reviews of selected transcripts were done by RW for quality control and to ensure reflexivity. Final transcriptions were transferred to Atlas software (Atlas.ti, Version 7 software, Berlin, Germany) for analysis. Thematic analysis was done using a predetermined frame with the following themes: [45, 46] willingness of HIV positive pregnant women to start ART, uptake of, readiness and motivation to initiate Option B+ ART, refusal and delay to take Option B+ ART, and early adherence to Option B+ ART as guided by the quantitative and qualitative tools [30]. We identified paragraphs linked to the predetermined themes and then coded these pieces inductively using thematic analysis. Thereafter, codes were grouped into subthemes within each pre-determined theme. Coding was done by AM. The codes and subthemes were discussed and reflected on by the co-authors through regular meetings. Distinctive quotes were identified and used to highlight the main issues identified in the interviews. This was done for both the in-depth and key informant interviews.

## Ethical considerations

Ethical approval was gotten from Makerere University School of Public Health Higher Degrees, Research and Ethics Committee and Uganda National Council for Science and Technology (UNCST). Authorization to conduct the study was sought from the district health officers and health facility in-charges. Written informed consent was attained from every study participant prior to conducting interviews. HIV positive pregnant women aged 15–17 years

who participated in the quantitative component of the study were considered as emancipated minors as per the UNCST ethics committee guidelines [47].

## Results

Both quantitative and qualitative results are integrated as per the pre-defined themes.

### Baseline characteristics

Out of 925 women who were screened for eligibility, 507 HIV positive expectant females were enrolled into the study. Fig 1 shows numbers of women who were screened, recruited in the study and included in the quantitative analyses at the first two-month follow-up visit.

Table 1 shows baseline characteristics of women who were enrolled in the cohort study (quantitative component). Mityana GH contributed the maximum number of study participants (42.1%, 213), followed by Masaka RRH (36.2%, 184) and Luwero (21.7%, 110).

The median (IQR) age was 24 (21, 28) years. Half (52.1%) were aged 15–24 years while 28.6% aged 25–29 years. Majority (51.1%) had primary or no education and 48.9% had attained secondary education and above. Four in five (80%) women were married, and Catholic was the most common religion (47.9%). A third (33.7%) of the women were involved in business/commercial activities, 23.1% were home-markers/ housewife, and 18.9% were engaged in subsistence farming. Nearly all study participants (95.5%) indicated that they intended to deliver from the study-enrolling health facility. Just over a quarter (27.2%) had been initiated on Option B+ ART within four weeks prior to enrolment into the study.

Forty-nine women were eligible to participate in the study but declined. Out of these, 41 women had data on their baseline characteristics as shown in Table 2. Majority (80.5%) were from Luwero HC IV, and the median (IQR) age was 24 (21, 27) years. More than half (53.7%) were aged 15–24 years while 31.7% aged 25–29 years. Most women (63.4%) had primary or no education and 36.6% had attained secondary education and above. Majority (82.9%) of women were married, and four in five (80.5%) were ART naive.

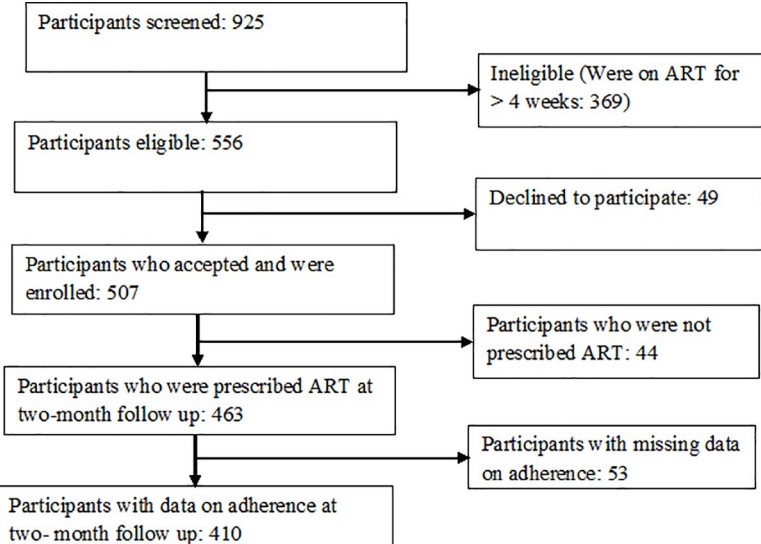

**Fig 1. Flow chart for enrolment and two- month follow-up of the study participants.**

**Table 1. Baseline characteristics of HIV positive pregnant women recruited into the study (N = 507).**

| Characteristic | Health Facility | | | |
|---|---|---|---|---|
| | Luwero HC IV (n = 110, 21.7%) | Mityana GH (n = 213, 42.1%) | Masaka RRH (n = 184, 36.2%) | Total number (%) |
| **Age (years)** | | | | |
| Median (IQR) | 23 (20–28) | 25 (21–29) | 24 (20.5–28) | 24 (21–28) |
| 15–24 | 63 (57.3) | 97 (45.5) | 104 (56.5) | 264 (52.1) |
| 25–29 | 29 (26.4) | 63 (29.6) | 53 (24.8) | 145 (28.6) |
| 30–44 | 18 (16.4) | 53 (24.9) | 27 (14.7) | 98 (19.3) |
| | Number (%) | Number (%) | Number (%) | |
| **Highest level of education completed** | | | | |
| Primary or below | 53 (48.2) | 123 (57.7) | 83 (45.1) | 259 (51.1) |
| Secondary or above | 57 (51.8) | 90 (42.3) | 101 (54.9) | 248 (48.9) |
| **Marital status** | | | | |
| Married | 89 (80.9) | 172 (80.8) | 145 (78.8) | 406 (80.1) |
| Never married | 12 (10.9) | 19 (8.9) | 30 (16.3) | 61 (12.0) |
| Widowed/separated | 9 (8.2) | 22 (10.3) | 9 (4.9) | 40 (7.9) |
| **Occupation** | | | | |
| Peasant farmer | 18 (9.8) | 24 (21.8) | 54 (25.4) | 96 (18.9) |
| Salaried | 10 (5.4) | 6 (5.5) | 13 (6.1) | 29 (5.7) |
| Business/commercial | 60 (32.6) | 24 (21.8) | 87 (40.9) | 171 (33.7) |
| Casual worker | 1 (0.5) | 10 (9.1) | 10 (4.7) | 21 (4.1) |
| Not employed | 34 (18.5) | 9 (8.2) | 0 (0.0) | 43 (8.5) |
| Housewife | 41 (22.3) | 28 (25.5) | 28 (22.5) | 117 (23.1) |
| Others | 20 (10.9) | 9 (8.2) | 1 (0.5) | 30 (5.9) |
| **Religion** | | | | |
| Catholic | 39 (35.5) | 99 (46.5) | 105 (57.1) | 243 (47.9) |
| Protestant | 35 (31.8) | 53 (24.9) | 20 (10.7) | 108 (21.3) |
| Born again | 17 (15.5) | 23 (10.8) | 15 (8.2) | 55 (10.8) |
| Muslim | 15 (13.6) | 28 (13.1) | 43 (23.4) | 86 (17.0) |
| Other | 4 (3.6) | 10 (4.7) | 1 (0.5) | 15 (3.0) |
| **Health Facility where the participant intends to deliver from** | | | | |
| Study site | 97 (88.2) | 209 (98.1) | 178 (96.7) | 484 (95.5) |
| Another facility | 13 (11.8) | 4 (1.9) | 6 (3.3) | 23 (4.5) |
| **ART Initiation status** | | | | |
| Within past 4 weeks | 37 (33.6) | 66 (31.0) | 35 (19.0) | 138 (27.2) |
| ART naive | 73 (66.4) | 147 (69.0) | 149 (81.0) | 369 (72.8) |

Women who were enrolled in the study and those who declined had similar baseline characteristics except for the categories of casual worker (p = 0.019) and housewife (p = 0.022) under the variable "Occupation" (Table 3).

Fifty-seven participants took part in the in-depth interviews. A total of 54 key informant interviews were conducted with 34 clinic staff category and 20 facility managers. Details of the in-depth and key informant interviews participants were described elsewhere [28, 36].

In the subsequent results, we present the quantitative results under the predetermined themes while qualitative results are under the subthemes shown in Table 4.

**Table 2. Baseline characteristics of HIV positive pregnant women who declined to participated in the study (N = 41) by health facility.**

| Characteristic | Health Facility | | | |
|---|---|---|---|---|
| | **Luwero HC IV** | **Mityana GH** | **Masaka RRH** | **Total number (%)** |
| | **(n = 33, 80.5%)** | **(n = 5, 12.2%)** | **(n = 3, 7.3%)** | |
| **Age (years)** | | | | |
| Median (IQR) | 24 (20–27) | 27 (22–29) | 20 (20–32) | 24 (20–27) |
| 15–24 | 18 (54.5) | 2 (40.0) | 2 (66.7) | 22 (53.7) |
| 25–29 | 11 (33.3) | 2 (40.0) | 0 (0.0) | 13 (31.7) |
| 30–44 | 4 (12.1) | 1 (20.0) | 1 (33.3) | 6 (14.6) |
| | Number (%) | Number (%) | Number (%) | |
| **Highest level of education completed** | | | | |
| Primary or below | 22 (66.7) | 2 (40.0) | 2 (66.7) | 26 (63.4) |
| Secondary or above | 11 (33.3) | 3 (60.0) | 1 (33.3) | 15 (36.6) |
| **Marital status** | | | | |
| Married | 28 (84.9) | 5 (100.0) | 1 (33.3) | 34 (82.9) |
| Never married | 2 (6.1) | 0 (0.00) | 2 (66.7) | 4 (9.8) |
| Widowed/separated | 3 (9.1) | 0 (0.00) | 0 (0.00) | 3 (7.3) |
| **Occupation** | | | | |
| Peasant farmer | 6 (18.2) | 1 (20.0) | 1 (33.3) | 8 (19.5) |
| Business/commercial | 7 (21.2) | 0 (0.00) | 1 (33.3) | 8 (19.5) |
| Casual worker | 5 (15.2) | 0 (0.00) | 0 (0.00) | 5 (12.2) |
| Not employed | 2 (6.1) | 0 (0.00) | 0 (0.0) | 2 (4.9) |
| Housewife | 12 (36.4) | 3 (60.0) | 1 (33.3) | 16 (39.0) |
| Others | 1 (3.03) | 1 (20.0) | 0 (0.0) | 2 (4.9) |
| **ART Initiation status** | | | | |
| Within past 4 weeks | 8 (24.2) | 0 (0.0) | 0 (0.0) | 8 (19.5) |
| ART naive | 25 (75.8) | 5 (100.0) | 3 (100.0) | 33 (80.5) |

## Prescription of ART by health providers

At enrolment into the study, 27.2% (138/507) of women had received ART prescription within 4 weeks, increasing to 91.3% (463/507) at the month-2 interview. Prescription of ART by month-2 was significantly lower in Masaka RRH (77.7%, 143/184, p<0.001) than Luwero HC IV (98.2%, 108/110) and Mityana GH (99.5%, 212/213).

## Willingness of HIV positive pregnant women to start ART

Women who had not been prescribed ART at month 2 (n = 44) were asked if they were willing to be initiated on ART using the quantitative tool. Of these, 39 (88.6%) were willing to be started on ART, four (9.1%) were not willing, and one participant (2.3%) had missing data. Reasons for not having been prescribed ART among those who wished to start included being told by health workers to go and think about it (71.8%, 28/39), not ready to start immediately after the HIV positive test (20.5%, 8/39), lack of ART at the previous ANC facility (2.6%, 1/39), wanted to come with spouse (2.6%, 1/39), and being uninformed of the presence of Option B + approach (2.6%, 1/39).

**Giving HIV positive pregnant women counseling and time to think about starting ART.** Similar to the quantitative results, many key informant and in-depth interview participants said that most HIV positive gravid females were counseled and given time to think about starting lifelong ART. Key informants envisioned that counseling and giving HIV positive

**Table 3. Comparison of baseline characteristics of HIV positive pregnant women recruited into the study (N = 507) and those who declined (N = 41).**

| Characteristic | Enrolled | Declined | P- value |
|---|---|---|---|
| | Total number (N = 507, %) | Total number (N = 41, %) | |
| **Age (years)** | | | |
| Median (IQR) | 24 (21–28) | 24 (20–27) | - |
| | Number (%) | Number (%) | |
| 15–24 | 264 (52.1) | 22 (53.7) | 0.844 |
| 25–29 | 145 (28.6) | 13 (31.7) | 0.673 |
| 30–44 | 98 (19.3) | 6 (14.6) | 0.460 |
| **Highest level of education completed** | | | |
| Primary or below | 259 (51.1) | 26 (63.4) | 0.129 |
| Secondary or above | 248 (48.9) | 15 (36.6) | 0.129 |
| **Marital status** | | | |
| Married | 406 (80.1) | 34 (82.9) | 0.664 |
| Never married | 61 (12.0) | 4 (9.8) | 0.675 |
| Widowed/separated | 40 (7.9) | 3 (7.3) | 0.891 |
| **Occupation** | | | |
| Peasant farmer | 96 (18.9) | 8 (19.5) | 0.925 |
| Salaried | 29 (5.7) | 0 (0.0) | 0.116 |
| Business/commercial | 171 (33.7) | 8 (19.5) | 0.062 |
| Casual worker | 21 (4.1) | 5 (12.2) | **0.019** |
| Not employed | 43 (8.5) | 2 (4.9) | 0.420 |
| Housewife | 117 (23.1) | 16 (39.0) | **0.022** |
| Others | 30 (5.9) | 2 (4.9) | 0.793 |
| **ARV Initiation status** | | | |
| Within past 4 weeks | 138 (27.2) | 8 (19.5) | 0.283 |
| ART naive | 369 (72.8) | 33 (80.5) | 0.283 |

**Table 4. Predetermined themes and subthemes.**

| Predetermined themes | Sub-themes |
|---|---|
| 1. Willingness of HIV positive pregnant women to start ART | a) Giving HIV positive women counseling and time to think about starting ART |
| | b) Desire to have an HIV negative baby |
| 2. Uptake of Option B+ ART | a) Adequate counseling |
| 3. Readiness to take Option B+ ART | a) More time and information |
| 4. Motivation to take Option B+ ART | a) Desire to remain healthy and deliver an HIV negative baby |
| | b) Perceived effectiveness of B+ ART |
| | c) Benefits of Option B+ ART to HIV negative spouses |
| 5. Refusal and delay to start taking Option B+ ART | a) Doubt of HIV positive result |
| | b) Fear to take ART |
| | c) Duration of ART |
| 6. Early adherence to Option B+ ART | a) Disclosure of HIV positive status |
| | b) Participation in health education talks and counseling |
| | c) Desire to remain healthy and deliver an HIV negative baby |
| | d) Busy schedule |
| | e) ART side effects |

expectant females time to think about Option B+ before prescription of ART could enhance acceptability and uptake of, and adherence to, Option B+ ART and retention in HIV care.

> *"When a woman tests HIV positive, we ask her to decide basing on the counseling she has received. If she says that she's not ready to start ART we allow her time to think and come back when ready"* Health provider, Hospital.

Congruently, many in-depth interview participants indicated that they were willing to initiate ART on the same day they tested HIV positive but the health providers told them to come back later after internalizing issues around lifelong ART.

> *"The health workers first found me with the virus (HIV positive), they told me to come back and get the medicines [ART] on another day. Even though I wanted it to be given to me on that very day they tested me, I had to come back later after thinking about it well"* IDI Participant, Mityana GH

**Desire to have an HIV negative baby.**   From the in-depth interviews, most participants indicated that they were willing to start on ART the same day they were found to be HIV positive so as to deliver an HIV negative baby. Desire for an HIV negative baby as the main reason for preference of same day ART initiation was also mentioned by key informant interview participants.

> *"After testing HIV positive, I was willing to start ART immediately but they [health workers] told me to go and think about it, then come back when ready to start on ART. I did not refuse to start immediately, because I was pregnant and I wanted to deliver a healthy [HIV negative] baby"* IDI Participant, Masaka RRH.

> *"The reason why they [women] are willing to test for HIV and start on Option B+ is because they are pregnant and want to prevent their babies from getting HIV infected"* Health provider, HC III

The quantitative results showed that three of the four women who were not willing to start ART were from Masaka RRH while the fourth was from Luwero HC IV. Fear of taking ART, being depressed due to the HIV positive status, and lack of a place for hiding ART from family members were the key reasons for not willing to start ART.

## Uptake of Option B+ ART

At two months follow up, the majority of the women had been prescribed ART and 93.3% (432/463) had started swallowing the medication (Table 5). Uptake of ART was higher in Masaka RRH and Mityana GH (95.8%) but lower in Luwero HC IV (85.2%). Most women started swallowing ART on the same day of prescription, (370/432; 85.7%), some started swallowing after 1–7 days, 52/432 (12.0%), and 10 (2.3%) started after one week.

**Adequate counseling.**   The qualitative component revealed that most key informants said that if the HIV positive gravid females were appropriately counselled, they would start swallowing ART on the same prescription day.

> *"Most of them [women] start swallowing ART on the same day of prescription once they are well counseled. The few who refuse to start on ART do so later after on-going counselling"* Health provider, HC III.

**Table 5. Enrolment into the study (N = 507), ART prescription (n = 463) and uptake of Option B+ (n = 432) by health facility.**

| Health Facility | Number enrolled | ART prescription/initiation (n, %) | Uptake of ART (Prescription and swallowing) [n, %] |
|---|---|---|---|
| Masaka RRH | 184 | 143 (77.7) | 137 (95.8) |
| Mityana GH | 213 | 212 (99.5) | 203 (95.8) |
| Luwero HC IV | 110 | 108 (98.2) | 92 (85.2) |
| **Total** | **507** | **463 (91.3)** | **432 (93.3)** |

Consistently, in-depth interviews s indicated that most women began swallowing their ART on the same day it was prescribed after being tested and counseled by health providers.

*"They [health providers] gave me medicines [ART] on the very day I tested HIV positive. I was well counseled and told to choose a particular time when to swallow those medicines at night. I decided to swallow at 9.00pm and I started immediately on that very day"* IDI Participant, Katikamu HC III.

*"I received good [meaning adequate] counseling from the health providers and I accepted to start on the medication [ART] there and then"* IDI Participant, Mityana GH.

## Readiness to take Option B+ ART

Women who started swallowing ART by two months' follow-up (n = 432) were requested to state their readiness to initiate ART at the time of prescription. Among women with data on this question, majority (82.9%, 310/374) were ready to immediately start, but 12.6% (47) wanted to delay while 4.6% (17) were not ready at all. Readiness to start swallowing ART immediately varied by health facility, Mityana GH (88.2%) and Masaka RRH (85.6%), but lower in Luwero HC IV (67.1%). Women were further asked to give reasons for not being ready to start ART at the time it was prescribed. Desire for more time (53.1%, 34/64), lack of sufficient information on Option B+ ART (23.4%, 15/64) and desire to repeat (confirmatory) HIV tests (9.4%, 6/64) were the most commonly mentioned hindrances to start taking ART. Other reasons cited are shown in Table 6.

**More time and information.** Similar to the quantitative findings, in-depth interview participants echoed the need for more time and information before prescription of ART. This was envisioned to enable one to understand how to take ART.

*"Health workers should first give us enough time and information before prescribing ART. If one is given the medicines [ART] before he/she has understood well all the necessary dos and*

**Table 6. Reasons for not being ready to start ART at the time of prescription (n = 64).**

| Reason | n (%) |
|---|---|
| Needed more time think about starting Option B+ | 34 (53.1) |
| Needed more information on Option B+ | 15 (23.4) |
| Wanted to do a repeat/confirmatory HIV test | 6 (9.4) |
| Wanted to first acquire a watch to set time for swallowing ART | 2 (3.1) |
| Wanted to first recover from sickness | 2 (3.1) |
| Lacked food | 1 (1.6) |
| Discouraged by a friend | 1 (1.6) |
| Wanted to deliver first | 1 (1.6) |
| Feared the size of the tablets | 1 (1.6) |
| Wanted to start after showing signs of HIV | 1 (1.6) |

*don'ts, one may end up taking it poorly or not taking ART at all"* IDI Participant, Mityana GH.

*"That time I wasn't given any information, because the health worker didn't explain well. I didn't get information on how to swallow the medicines and care for myself"* IDI Participant, Masaka RRH.

## Motivation to take Option B+ ART

The quantitative results showed that motivation to start taking Option B+ ART by women at all facilities was majorly for own health (92.3%, 334/362), and to protect their unborn babies (90.6%, 326/362) or spouses (7.5%, 27/362) from HIV infection. Additional reasons that were obtained through the quantitative interviews include; advise from health providers, falling sick frequently, desire to live longer, fear of symptoms of HIV disease to be noticed and encouragement from spouses.

**Desire to remain healthy and deliver an HIV negative baby.**   Similar to the quantitative results, in-depth interviews revealed the desire to stay healthy and have an HIV negative baby by women as a motivation to start Option B+ ART. Participants indicated that they needed to be healthy to continue working and look after their families. Most participants felt that when women are given adequate counseling and told the benefits of Option B+ ART, with the majority accepting ART.

*"I wanted to save my life, remain healthy so that I can look after my family and deliver a healthy [HIV negative] child too"* IDI Participant, Masaka RRH.

Key informant interview participants also alluded to desire for an HIV negative baby as a motivation to take Option B+ ART.

*"When these mothers are tested and they are found to be HIV positive, they are advised to start on the medicines [ART]. Most women accept to take the medicines [ART] because they want to give birth to an HIV negative child"* Health provider, HC IV.

**Perceived effectiveness of Option B+ ART.**   All in-depth interview participants who were good ART adherers emphasized that if ART is taken well, they make one healthy and the baby is born HIV negative. This was attributed to the effectiveness of Option B+ ART in comparison to the strategies before the Option B+ era and motivated them to take the ART.

*"The good thing I see in this one they brought (meaning Option B+), is that it seems better than the old one which came first [Option A] because most of the time, when women deliver, their children don't have the sickness (meaning HIV negative babies), unlike in the past, some could deliver, and when you test, they could find the babies infected with the virus"* IDI Participant, Mityana GH.

Furthermore, good ART adherers illustrated the experience of women that once on ART, one lives longer than those who were not taking ART. It was noted that this enables HIV positive women to care for their children for a long time.

*"Medicines [meaning ART] help us to live longer and look after our children for a number of years"* IDI Participant, Katikamu HC III.

**Benefits of Option B+ ART to HIV negative spouses.** Most in-depth interviews revealed that participants were either not aware of the benefits of the Option B+ ART to their spouses or were only interested in the benefits for the woman's health and that of her baby.

> "*They [health providers] told me that if I concentrate on taking the ART, the baby will not be infected with HIV and I will remain health. I was not told anything in regards to the benefits of ART to my spouse*" IDI Participant, Mityana GH.

## Refusal and delay to start taking Option B+ ART

Quantitative results revealed that thirty-one participants who received a prescription and ART had not yet started taking their ART by two months of the study. Reasons cited were; not ready/ prepared (14), fear to take ART for life (9), preference for local herbs due to fear of ART tablets (3), wanting to start taking after giving birth (3), and doubting the HIV positive result (2) and wanting to have a repeat (confirmatory) HIV test.

Both providers and women alluded to similar reasons during qualitative interviews as per the sub-themes below.

**Doubt of HIV positive results.** Some women who delayed to start taking ART indeed said that if they are given a positive HIV result for the first time, they would go to other health facilities to repeat the test. The women said they would start taking ART after getting a confirmatory HIV positive test.

> "*When I got the first pregnancy, I was told that I was HIV positive. However, I never believed, but took Septrin tablets only. When I got the second pregnancy, I had not yet started on the big tablets [ART]; I tested for HIV again wanting to prove whether I was truly positive. The test gave the same results [HIV positive]. I had nothing to do but to start taking ART*" IDI Participant, Masaka RRH.

Likewise, key informant interview participants indicated that some women doubt positive HIV results and request for repeat tests.

> "*Some women refuse, 'am not HIV positive*!' *If we test a woman and she doubts the results we have given her, we repeat the test or refer her to another Health facility 'X'—they believe in it so much. So, we refer them there and wait for the results.*" Health provider, HC IV.

**Fear to take ART.** Among women who participated in in-depth interviews, fear to take ART was identified as a barrier to start ART. Women expressed a number of fears like taking ART for life, being seen by the spouse, and possible side effects.

> "*I delayed to start the ART because I was fearing. I feared to swallow it all the time and I kept thinking of the medicines [ART] all the time*! *But now am no longer thinking like that*". IDI Participant, Masaka RRH.

> "*I did not start immediately because I had fear, I had to think for some time, like a number of days. I first feared, because people scared us that, the tablet is very big and things like that*" IDI Participant, Masaka RRH.

> "*At first I feared to start swallowing the ART because I did not want the man [spouse] to see me taking medicines. He had told me that, if they test you and you are found with the HIV virus, we will separate [meaning divorce]*" IDI Participant, Mityana GH,

A few women had a misconception that some people can even die faster if they start swallowing ART. The community was said to be the source of such misinformation.

*"People in the village (those who were swallowing ART), told us that there is that new ART (Option B+ ART) which kills people very fast- some people come when they are sick [have advanced HIV] and they are given what kills fast."* IDI Participant, Ssunga HC III.

**Duration of ART.**    A number of women indicated that they were scared of taking ART for the rest of their lives. They went ahead to compare taking ART to other short course medicines such anti-malaria medicines, which take only a few days.

*"I was worried, because I didn't know how to swallow the daily medicines [ART] for life. I am used to short treatment: when I am sick of malaria, I swallow two tablets and I know that I am now healed, but I thought of the daily medicines [ART]. . . ..will I manage, I was worried!"* IDI Participant, Katikamu HC III.

## Early adherence to Option B+ ART

Adherence was assessed on 410 participants who had complete quantitative data at the two-month follow up. Just over three quarters (76.8%, 315/410) of study participants had optimal ART adherence. Optimal adherence was more common among women who were ready to immediately start ART at time of prescription 83.2% (258/310), and lower for women who wished to delay the start of ART 64.1% (25/39), and lowest if a woman did not want to ever start, 17.6% (3/17). Table 7 shows the reasons given for non-adherence. Most common reasons from the quantitative interviews included having travelled far away (24.2%, 23/95), side effects of ART (16.8%, 16/95), running out of ART (14.7%, 14/95), and forgetting to take ART (12.6%, 12/95).

Table 8 shows factors associated with optimal early adherence to Option B+ ART. Prevalence ratios are indicated, and no interactions were found among factors analysed. Optimal ART adherence was similar among women at Mityana GH and Masaka RRH, which were significantly higher than at Luwero HC-IV. Other factors associated with optimal adherence were readiness to immediately start and disclosure of HIV status to anyone (Table 8).

**Disclosure of HIV positive status.**    In the qualitative interviews, disclosure of HIV positive status was mentioned both in in-depth and key informant interviews as a facilitator of

**Table 7. Reasons for non-adherence to Option B+ ART (n = 95).**

| Reason | n (%) |
|---|---|
| Travelled far and failed to pick ART | 23 (24.2) |
| Side effects of ART | 16 (16.8) |
| Ran out of ART | 14 (14.7) |
| Forgot to swallow ART | 12 (12.6) |
| Lacked food to eat before swallowing ART | 06 (6.3) |
| Non-disclosure of HIV status to partner | 06 (6.3) |
| Had no money for transport to pick ART | 05 (5.3) |
| Doubt of HIV positive status | 04 (4.2) |
| Too sick to pick ART | 04 (4.2) |
| Feared the tablets due the big size | 03 (3.2) |
| Lost the bag which contained the ART | 01 (1.1) |
| Was on many other medicines | 01 (1.1) |

optimal adherence. Women who disclose their HIV positive status to their partners or a close relative get reinforced to continue taking their ART. However, those who don't disclose their status end up hiding their ART for fear of being seen by their spouses.

> *"He [spouse] has helped me a lot, it's very important to disclose. Those who didn't disclose have faced problems; they hide the medicines [ART] so that the spouse cannot see them and they end up missing some doses"* IDI Participant, Luwero HC IV.

**Participation in health education and counseling.**   In-depth interview participants who were given health education and satisfactory counseling stated that this helped them to start and remain adherent to the ART.

> *"The health workers first counseled me. I was told to start the medicines [ART] and from then up to today I have never missed. They [health workers] told me to take it [ART] on time without missing, I started swallowing without missing, and even I swallow at the exact hour/time. I have never missed even a single day".* IDI Participant, Masaka RRH.

On the other hand, lack of health education and counseling were highlighted as barriers to starting and remaining adherent to Option B+ ART. Some of the women who had suboptimal ART adherence attributed it to having not attended the pre-ART health education talks and counseling sessions.

> *"I used to reach when they had already given health education to other women. Therefore, I was not provided with any pre-ART education and orientation sessions before I started taking ART"* IDI Participant, Kyanamukaka HC IV.

> *"Mmhmmh. . .they just gave me the tin [meaning the tin containing ART] and I took it home to swallow. I almost I did not get any health education at all. I shouldn't lie to you"* IDI Participant, Mityana GH.

Equally, most key informant interview participants stated that good counseling could result in adherence to clinic appointments and ART.

> *"For me, I think that if the counseling has been good to the extent that the mother gets to know the benefits of option B+, I believe, she can adhere to her appointments and even take her medicines well for reasons that she wants an HIV negative child"* Health Provider, Luwero HC IV

**Desire to remain healthy and deliver an HIV negative baby.**   Most in-depth interview participants said that they didn't want to miss any ART doses so as to remain healthy and give birth to HIV negative babies. They anticipated that being adherent to the ART would result in a rise in their CD4 count and thus remain healthy.

> *"I can't miss any of my doses because I want my "asikaris" [meaning CD4 cells] to go higher all the time to remain healthy, so that am able to deliver a healthy child [meaning HIV negative]"* IDI Participant, Mityana GH.

**Busy schedule.**   During the in-depth interviews, some participants attributed suboptimal adherence to ART to busy schedules. A number of women had many chores to accomplish which resulted in their forgetting to swallow ART.

**Table 8. Factors associated with early adherence to Option B+ ART (n = 410).**

| Factors | Unadjusted PR | 95%CI | Adjusted PR** | 95%CI |
|---|---|---|---|---|
| **Health facility** | | | | |
| Masaka RHH | 1.66 | 1.30, 2.12 | 1.33 | 1.04, 1.69 |
| Mityana GH | 1.86 | 1.48, 2.35 | 1.44 | 1.14, 1.81 |
| Luwero HCIV | 1 | 1 | 1 | Referent |
| **Disclosure of HIV status to anyone** | | | | |
| Yes | 1.51 | 1.22, 1.87 | 1.24 | 1.04, 1.48 |
| No | 1 | 1 | 1 | Referent |
| **Motivation for starting ART: Protect baby** | | | | |
| Yes | 1.58 | 1.14, 2.19 | | |
| No | 1 | 1 | | Referent |
| **Motivation for starting ART: Protect self** | | | | |
| Yes | 1.90 | 1.23, 2.92 | | |
| No | 1 | 1 | | Referent |
| **Readiness to start ART** | | | | |
| Immediately | 1.67 | 1.27, 2.17 | 3.22 | 1.15, 8.96 |
| Later/Not at all | 14.73 | 1 | 1 | Referent |

Note**: Adjusted for health facility, HIV disclosure status, motivation to start ART and readiness to start ART.

*"At times you become too busy with many things like house work and digging. Time passes and that's when you forget to swallow the medicines [ART]"* IDI Participant, Kyanamukaka HC IV.

**ART side effects.** Side effects due to ART were commonly mentioned as being responsible for suboptimal adherence. In-depth interview participants mentioned varying side effects including dizziness, headache, drowsiness, nausea, vomiting and nightmares.

*"At the start, I felt bad whenever I would swallow the ART. I was vomiting, all the time I would feel like vomiting, just like someone suffering from malaria"* IDI Participant, Katikamu HC III.

## Discussion

This study assessed uptake and early adherence to ART and associated factors among HIV positive gravid and lactating mothers starting lifelong ART in Central Uganda. Results from this study show that majority (91.3%) of expectant HIV positive women were prescribed ART immediately after they tested positive, as recommended by the MOH. Our finding is slightly lower than that of Uganda population-based HIV impact assessment (UPHIA)-2016/2017 where 95.3% of HIV positive mothers who were 12 months post-delivery reported receiving ART. The slight difference might be ascribed to the fact that the UPHIA involved women who were on ART at the time of their first ANC and had delivered 12 months prior to the survey [48].

Most women who had not received a prescription in our study were willing to be started on ART, which indicates a missed opportunity for immediate ART initiation. Uptake of Option B + ART (prescription and swallowing of ART) was nearly universal by two months of follow-up, which is very encouraging. Importantly, most women started taking their ART on the same day of prescription. However, prescription and uptake of ART varied by health facility

which might be attributed to differences in practices such as assessing women for readiness to start ART, giving women time to get ready, providing adequate counseling, and ensuring that women understood the benefits of Option B+ ART. Whereas it is desirable to start ART immediately, women who are not ready to start treatment should be screened and provided with additional follow-up support. Further, the delay to start taking the medicines after prescription highlights the need to carefully track the immediate start and to support women who may encounter challenges in this early phase. This is congruent with recommendations by a trial conducted at 70 research sites in 15 countries within sub-Saharan Africa, Asia, and the Americas where one-third of asymptomatic HIV positive pregnant and postpartum women did not initiate ART immediately for desire for more time to make a decision [49].

Initiation of Option B+ ART by health providers and actual taking in this study was higher than reported elsewhere [11, 12, 16]. The high level of ART uptake could be due to the wish to have HIV free infants as well as the women's own health status. For the barriers such as preference for local herbs, fear of ART side effects, among others reflect a need for further education and support including counseling for reassurance and to reduce misinformation.

Early adherence was suboptimal (76.8%), although uptake was very high and most women were ready to start Option B+ ART immediately and knew the reasons for taking ART. Such a low adherence level soon after ART initiation is worrying since women are expected to be more motivated to take their medicines regularly at this stage than later on in the PMTCT cascade [18, 50]. Attention to the prominent reasons for this, including non-disclosure of HIV positive status, side effects of ART, and perception that the baby is safe post-delivery might be useful [28]. Ensuring that women are ready for Option B+ and HIV positive status disclosure are factors that can be stressed to improve lifelong ART adherence. HIV positive gravid women should not be hurried to start on Option B+ ART on the same day of HIV diagnosis but be sufficiently counseled and supported to ensure that they are ready for the lifelong ART. Facilities should have mechanisms such as use of peers to detect women who are ART non-adherent early enough so that they can be given targeted supportive adherence counseling [24]. These peers should be well-trained to acquire knowledge and skills necessary for detecting non-adherence. Adequate adherence results into viral suppression and subsequently less risk of MTCT of HIV [51] as endorsed in the UNAIDS concept of undetectable = untransmittable [52].

Most of the issues around ART prescription, swallowing and early adherence among HIV positive gravid and lactating women that were identified in this study are critical and can be addressed by strengthening the existing strategies. Such strengthening would focus on provision of adequate general and targeted counseling, and social support and early tracking of women. Although some study eligible women declined to participate in the study, their baseline characteristics did not vary from those who participated except for the categories of casual worker and housewife under the variable "Occupation". Although some earlier studies found a positive association between occupation/employment and adherence [53], similar to our study, other studies reported no association between socio- demographic and economic variables and adherence to ART [54, 55]. Therefore, our study findings can be generalizable to women in settings similar to where the study was conducted such as regional referral hospitals, general hospitals and HC IIIs.

## Implications for practice, public health and policy

Our study results have implications for practice, public health and policy. These mainly revolve around expectant females and lactating mothers, their close networks and health providers.

Understanding the benefits of lifelong ART, early ART initiation, optimal ART adherence, and HIV status disclosure could heighten uptake of and adherence to ART [56, 57]. Addressing barriers to non-adherence such as long distances to health facilities, fear of side effects, non-disclosure of HIV status to anyone, and perception that the baby is safe from HIV infection post-delivery is crucial [58, 59] and calls for more time and adequate information during PMTCT health education and counselling. Additionally, health workers should be cognizant of challenges women face with uptake and adherence, such as unwillingness and unreadiness to start lifelong ART, as well as stigma which deters disclosure of HIV positive results, and should work to adequately counsel and support them.

Ministry of Health should institute appropriate pre-and in-service training, guidelines, supervision, and mentorship for health workers to enhance quality PMTCT counseling. Furthermore, inadequate counselling due to lack of time and health workers could be addressed through strategies such as task shifting using peers to address human resource gaps. Finally, women who are not ready for lifelong ART should be given time and support to accept newly diagnosed HIV positive status before ART initiation.

## Strengths and limitations

To our knowledge, this is among the first studies that has documented detailed day-to-day critical steps in the preparation and ART initiation such as prescription, uptake, readiness, motivation and early adherence of Option B+ medicines among pregnant women. The study used data from a prospective cohort of HIV positive expectant females complemented with data from qualitative interviews. Use of the integrated mixed methods study approach facilitated greater understanding of issues around prescription, swallowing, and uptake of, and adherence to lifelong ART among expectant and lactating women. Unfortunately, uptake and adherence were assessed by self-report which could have introduced social desirability bias. However, the two outcome variables were measured using a series of questions. In addition previous studies used self-report while measuring uptake and adherence [22, 60, 61]. Our study had a limitation of recall bias since we used a 30-days recall method to assess adherence. Nevertheless, prior studies have used the same technique to measure ART adherence [61–63]. Using individual interviews might have resulted into social desirability bias but this was minimized by use of well-trained and experienced data collectors, establishing of rapport, use of probes, conducting regular debriefing sessions, and research team meetings [64].

## Conclusions

Most women received ART prescription on the same day they were diagnosed HIV positive. Uptake of Option B+ in the first two months after HIV counseling, testing and ART initiation among females seeking ANC service is high in this setting. However, early adherence to Option B+ ART remains suboptimal especially in the lower facility. Disclosure of HIV status and readiness to start ART at time of prescription, a reflection of quality of counseling are important determinants of optimal adherence. Women still require time to internalize or accept the newly learnt HIV status and to be provided with more information prior to starting Option B+ ART.

## Supporting information

**S1 Appendix. Key informant interview guide.**
(PDF)

**S2 Appendix. In-depth interview guide.**
(PDF)

## Acknowledgments

The authors appreciate the study participants and the respective in-charges of Masaka RRH, Mityana GH, Luwero HC IV, Kyanamukaka HC IV, and Ssunga and Katikamu HC IIIs for their sustained support given to us throughout the study period. Additionally, special thanks go to the district health office of Masaka, Mityana, and Luwero that provided us with the necessary support to conduct the study. We appreciate the research assistants who carried out data collection and transcription. Special thanks also go to Mr. Edward Were and Mr. Ronald Ssenyonga who supported us in data management and analysis.

## Author Contributions

**Conceptualization:** Aggrey David Mukose, Fredrick Makumbi, Esther Buregyeya, Joshua Musinguzi, Rhoda K. Wanyenze.

**Formal analysis:** Aggrey David Mukose, Hilde Bastiaens, Fredrick Makumbi, Rhoda K. Wanyenze.

**Funding acquisition:** Joshua Musinguzi, Rhoda K. Wanyenze.

**Investigation:** Aggrey David Mukose, Fredrick Makumbi, Esther Buregyeya, Joshua Musinguzi, Rhoda K. Wanyenze.

**Methodology:** Aggrey David Mukose, Fredrick Makumbi, Esther Buregyeya, Joshua Musinguzi, Rhoda K. Wanyenze.

**Project administration:** Rose Naigino.

**Supervision:** Hilde Bastiaens, Fredrick Makumbi, Jean-Pierre Van Geertruyden, Rhoda K. Wanyenze.

**Validation:** Aggrey David Mukose, Rhoda K. Wanyenze.

**Visualization:** Aggrey David Mukose, Hilde Bastiaens, Fredrick Makumbi.

**Writing – original draft:** Aggrey David Mukose.

**Writing – review & editing:** Aggrey David Mukose, Hilde Bastiaens, Fredrick Makumbi, Esther Buregyeya, Rose Naigino, Joshua Musinguzi, Jean-Pierre Van Geertruyden, Rhoda K. Wanyenze.

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
