## [Decision Letter · Decision Letter 0]

8 Jan 2021

PONE-D-20-22840

What influences uptake and early adherence to Option B+ medicines among HIV positive pregnant and breastfeeding women in Central Uganda? A mixed methods study

PLOS ONE

Dear Dr. Mukose,

Thank you for submitting your manuscript to PLOS ONE. After careful consideration, we feel that it has merit but does not fully meet PLOS ONE’s publication criteria as it currently stands. Therefore, we invite you to submit a revised version of the manuscript that addresses the points raised during the review process.

Two reviewers have made very specific recommendations for revising the manuscript, and each of these should be fully addressed if you wish to submit a revised version.

We look forward to receiving your revised manuscript.

Kind regards,

Julie AE Nelson, PhD

Academic Editor

PLOS ONE

Journal Requirements:

3. During our internal review, we noticed that you cited one of your own published studies that presents similar qualitative data (published in BMC Preg & Childbirth in 2017). Please provide further clarification on how your current qualitative findings advance on your previously published study.

Reviewers' comments:

Reviewer's Responses to Questions

**Comments to the Author**

1. Is the manuscript technically sound, and do the data support the conclusions?

Reviewer #1: Yes

Reviewer #2: Yes

2. Has the statistical analysis been performed appropriately and rigorously? 

Reviewer #1: Yes

Reviewer #2: Yes

3. Have the authors made all data underlying the findings in their manuscript fully available?

Reviewer #1: Yes

Reviewer #2: Yes

4. Is the manuscript presented in an intelligible fashion and written in standard English?

Reviewer #1: Yes

Reviewer #2: Yes

5. Review Comments to the Author

Reviewer #1: Antiretroviral therapy (ART) adherence is an important subject in sub-Saharan Africa especially among pregnant and breastfeeding women. The persistence of new HIV infections among new paediatrics is a concern. Thus, this topic is relevant, and authors should be commended for undertaking this study.

General comments. I will strongly recommend that the word ARVs be substituted throughout the manuscript with ART. In the context of this research where a fixed dose combination of three active drugs is in use, ART will be the appropriate term to use. In addition, the use of good adherence suggests that there is better and best adherence. It will be appropriate to use optimal adherence and suboptimal adherence where applicable in the manuscript.

Study sites

1. Authors should explain what high patient volume and low patient volume facilities entails.

Quantitative component

1. ‘’Enrolment was done concurrently at all the three sites until the desired sample size (estimated at 500) was achieved.’’ Please explain how the study size was arrived at.

2. The pre-tested and standardized questionnaires. Can the author briefly state how this was done?

3. Quantitative data collection for this paper (change paper to study) took place between October 2013 and October 2014.

Qualitative component

1. Were field notes made during and/or after the interviews? If yes, were they reflected in the results section?

2. What was the duration of the interviews?

3. Data saturation was not considered in this study. Why was data saturation not considered during data collection?

4. Were repeat interviews carried out? If yes, how many?

5. Five of the lead investigators (A.M., R.W., E.B., R.N., and F.M.) interviewed the KIs: please spell KIs in full. Limit the use of abbreviations to the key terms in the study. KIIs: please spell in full. This is confusing to have KIs and KIIs. Just spell them in full.

6. ‘’The specific questions to the key informants were ……’’ Please include interview schedule used in this study as appendix to the manuscript.

7. ‘’The questions to the IDI participants included;….’’ this level of information is not required in the manuscript. Please include interview schedule used in this study as appendix to the manuscript.

Data collection tools and measures

1. ‘’……. written informed consent to the eligible women, and subsequently conducted face-to-face interviews using structured pre-tested and standardized questionnaires. Can the author briefly state how this was done?

Results

1. Overall, in the three facilities where the quantitative study was conducted, how many women were eligible to participate in this study, how many declined to participate and how many eventually participated?

Discussion

1. ‘’HIV positive pregnant women should not be rushed to start on Option B+ ARVs on the same day of HIV diagnosis but be prepared well.’’ Can the authors elaborate on what they mean by ‘’ but be prepared well?’’

2. ‘’Facilities should have mechanisms to detect women who are nonadherent

3. early enough so that they can be given supportive adherence counseling [24].’’ Authors should discuss a few of these mechanisms.

Strength and Limitations

1. Are there any other limitations, perhaps associated with the methodology (e.g. Individual interviews, etc)?

2. A 30 days recall method was used to assess adherence. Recall bias was not mentioned as your limitation?

Conclusion and recommendation.

1. The authors should have a separate section for conclusion and recommendation. In addition, authors should discuss/provide the implications of their findings for practice, public health and policy.

Others

1. Authors should discuss the generalisability (external validity) of the study results.

Reviewer #2: Well written manuscript with adequate background information about the problem, study gap or rationale, and study objectives. Appropriate methods as regards study design, selection criteria, setting, data sources, measurements, operational definitions, human subject and ethical considerations, and analysis were articulated.

However, few details may need to be addressed to improve clarity and interpretation findings of the manuscript:

Methods:

1. The authors referenced parent study for sample size. However, it is critical for the authors to have demonstrated whether the parent sample size (references 28 and 29) was adequate in precision or power to detect the desired outcomes for the findings in this manuscript – uptake, early adherence and associated factors. Assumptions and power calculations could have been stated.

2. In the analytic strategy, it was important for the authors to have stated whether they explored for interaction or effect modification for certain factors on to the outcome, particularly facility level interacting with other factors. Participants and factors at health center IV could have been different at general and referral hospitals.

Results: The results are relevant to the problem posed. Synthesis of results was satisfactory. However, the presentation may need to be improved.

1. A table showing how comparable the select baseline variables in Table 1 from participants who declined (49 in figure 1) versus those who were included in the study. This would support interpretations on external validity of the findings.

2. Table 6 shows results with appropriate analytic strategy; however, certain baseline variables in Table 1 were not considered. Further, and as already discussed by the authors that uptake and adherence may differ by health facility level, authors could have explored possible interactions between health facility level versus certain baseline variables and variables in Table 6 onto the outcome. Findings from exploring interactions would support the need for certain targeted interventions at different levels of the health facility as compared to the general recommendations that have been articulated in the discussion.

Interpretation and conclusions: The discussion was balanced with articulation of the limitations and strengths. The explanations or theories were appropriate to the claims.

1. However, the authors may need to revise the discussion when results change in line with the suggestions I have made in the methods and result sections.

2. The limits of the external validity or generalizability were not explicitly discussed

6. PLOS authors have the option to publish the peer review history of their article (what does this mean?). If published, this will include your full peer review and any attached files.

Reviewer #1: No

Reviewer #2: **Yes: **Ezekiel Mupere MBChB, MMed, MS., PhD

---

## [Author Response · Author response to Decision Letter 0]

7 Mar 2021

PONE-D-20-22840

What influences uptake and early adherence to Option B+ medicines among HIV positive pregnant and breastfeeding women in Central Uganda? A mixed methods study

PLOS ONE

Dear Dr. Mukose,

Thank you for submitting your manuscript to PLOS ONE. After careful consideration, we feel that it has merit but does not fully meet PLOS ONE’s publication criteria as it currently stands. Therefore, we invite you to submit a revised version of the manuscript that addresses the points raised during the review process.

Two reviewers have made very specific recommendations for revising the manuscript, and each of these should be fully addressed if you wish to submit a revised version.

Response to comments: 

Dear Reviewers and the Academic Editor, we thank you for your time. We also thank you for the helpful comments to improve our manuscript. We have responded to all the concerns and accordingly revised the manuscript. We have addressed the additional requirements and each of the reviewers’ recommendations. Please find detailed point by point responses to reviewer comments below and the changes highlighted in the “Revised Manuscript with Track Changes”.

We look forward to receiving your revised manuscript.

Kind regards,

Julie AE Nelson, PhD

Academic Editor

PLOS ONE

Journal Requirements:

Thank you so much for asking for more information on the study questionnaire.

The questionnaire used for data collection was adopted with modifications. We have included the references. The key measurements included socio-demographic characteristics, prescription and swallowing of Option B+ ART, reasons for non-prescription and refusal to swallow ART, disclosure of HIV status, willingness to start ART, readiness to start taking (initiate) ART, motivation to initiate ART, reasons for delay to initiate ART, understanding of how long the participant was supposed to be on ART and self-reported adherence to ART. Lines 141 and 142 on page 8, then lines 193-199 on page 10 in the revised manuscript with track changes. 

3. During our internal review, we noticed that you cited one of your own published studies that presents similar qualitative data (published in BMC Preg & Childbirth in 2017). Please provide further clarification on how your current qualitative findings advance on your previously published study.

Thank you so much for the internal review comment. Our current qualitative study findings advance on our earlier paper published in BMC Preg & Childbirth in 2017 in the following aspects.

1. Mixed methods were used in the current manuscript to ensure complementarity and facilitation of greater understanding of issues around prescription of, uptake and early adherence to ART

2. The qualitative arm in this study/manuscript explored experiences around lifelong ART prescription (Being given time to think, willingness, doubt of positive HIV results, readiness, refusal and delay to start ART, and adequate counselling), uptake and early adherence among both women on Option B+ and perspectives of Option B+ service health providers. The earlier paper focused on facilitators and barriers to uptake and adherence from the HIV positive women’s perspective.

We thank you so much Academic Editor for noting issues around data availability.

Patients were not consented to provide data access and the Higher Degrees, Research and Ethics Committee didn’t grant permission for data access. We have revised the data availability statement and provided the contact address for the Chair of the Higher Degrees, Research and Ethics Committee. The data availability statement should now read as below:

“Due to restrictions by the Makerere University School of Public Health Higher Degrees Research and Ethics Committee, some access restrictions apply to the data for reasons of safety and protection of study subjects and their institutions. Sensitive data was collected from patients and they didn’t consent to open data access. However, criteria eligible researchers with interest in the data may request for anonymized data access through the Chair Higher Degrees, Research and Ethics Committee.”

Thank you for the suggestion. The contact information for the ethics committee is given below. 

The Chairperson Makerere University School of Public Health Higher Degrees, Research and Ethics Committee, P.O. Box 7072, Kampala. Telephone +256414 532207/543872/543437

Dear Academic Editor, there are restrictions as noted above. 

Thank you so much. Kindly revise the Data Availability Statement on our behalf to read as stated under number 4 above.

Reviewers' comments:

Reviewer's Responses to Questions

Comments to the Author

1. Is the manuscript technically sound, and do the data support the conclusions?

Reviewer #1: Yes

Reviewer #2: Yes

2. Has the statistical analysis been performed appropriately and rigorously?

Reviewer #1: Yes

Reviewer #2: Yes

3. Have the authors made all data underlying the findings in their manuscript fully available?

Reviewer #1: Yes

Reviewer #2: Yes

4. Is the manuscript presented in an intelligible fashion and written in standard English?

Reviewer #1: Yes

Reviewer #2: Yes

5. Review Comments to the Author

 Reviewer #1: Antiretroviral therapy (ART) adherence is an important subject in sub-Saharan Africa especially among pregnant and breastfeeding women. The persistence of new HIV infections among new paediatrics is a concern. Thus, this topic is relevant, and authors should be commended for undertaking this study.

Thank you so much for the kind and encouraging remarks.

General comments. I will strongly recommend that the word ARVs be substituted throughout the manuscript with ART. In the context of this research where a fixed dose combination of three active drugs is in use, ART will be the appropriate term to use. In addition, the use of good adherence suggests that there is better and best adherence. It will be appropriate to use optimal adherence and suboptimal adherence where applicable in the manuscript.

We thank you for the general comments. We have changed from the word ARVs to ART throughout the manuscript. We have also used the terms optimal and suboptimal adherence as suggested.

Study sites

1. Authors should explain what high patient volume and low patient volume facilities entails.

According to the Uganda Ministry of Health guidelines, high patient volume facilities have more than 500 patients in HIV care whereas low volume facilities have less than 500 patients in HIV care. We have included a reference (MOH. Consolidated Guidelines for Prevention and Treatment of HIV in Uganda. In: ACP, editor. Kampala, Uganda: MOH; 2016. p. 154). Please see lines 125-126, page 7.

Quantitative component

1. ‘’Enrolment was done concurrently at all the three sites until the desired sample size (estimated at 500) was achieved.’’ Please explain how the study size was arrived at.

This was the sample size of 500 was for the parent study and was determined as described in our earlier paper by Naigino, R., Makumbi, F., Mukose, A., Buregyeya, E., Arinaitwe, J., Musinguzi, J. and Wanyenze, R.K., 2017. HIV status disclosure and associated outcomes among pregnant women enrolled in antiretroviral therapy in Uganda: a mixed methods study. Reproductive Health, 14(1), p.107. 

In order to estimate the sample size needed for the study primary objective, enrolment into Lifelong ART by all diagnosed HIV+ women during ANC, the following assumptions were made. The assumed proportion of HIV+ women who would enrol was p=80%. This was based on the 75% enrolment into PMTCT of HIV+ pregnant women. However, the assumed 80% was arrived at because some of the key factors associated with lower enrolment such as the tedious and completed assessment of who is eligible for PMTCT would have been eliminated because all diagnosed HIV+ are eligible for lifelong ART, irrespective of their CD4 or WHO clinical stage. However, adherence to counseling sessions and prolonged waiting times would still be important barriers to returning for PMTCT enrolment, resulting into only 5% change. Other key assumptions were a two-sided α= 0.05, and a 5% level of precision of the estimate thus resulting in 246 as sample size. Adjusting for design effect due clustering of 2 because two levels (district and HC) who participated in this study, the sample size came to 496 (~500). Due to concurrent enrolment at all the 3 sites, a total of 507 were recruited. 

We have included some of this information and put this reference. Kindly see lines 147-149 on page 8.

2. The pre-tested and standardized questionnaires. Can the author briefly state how this was done?

After training the research assistants, pre-testing of the questionnaires was done by administering them to a sample of study eligible participants in a non-study site. Thereafter, the investigators met with the research assistants to go through the entire tool to correct any unclear questions. This ensured that all participants were asked the same questions in an identical format and responses recorded in a uniform manner. Please see lines 142-146, page 8.

3. Quantitative data collection for this paper (change paper to study) took place between October 2013 and October 2014.

Thank you for the suggestion, we have changed from “paper” to “study” as shown in line 161 on page 8.

Qualitative component

1. Were field notes made during and/or after the interviews? If yes, were they reflected in the results section?

Dear Reviewer, field notes were taken alongside the audio recording during the interviews. They were used during transcription and are therefore reflected in some of the sub-themes and quotes.

2. What was the duration of the interviews? 

Thank you so much Reviewer. We have included the duration of each key informant and in-depth interview. Kindly see lines 189-190, on page 10.

3. Data saturation was not considered in this study. Why was data saturation not considered during data collection?

Thank you so much for the question. We had an estimated sample size before data collection other than considering data saturation as explained below. This was necessary in our research context because sample size specification and justification were required by the funder, ethics committee, and Ministry of Health the study was implemented. This method is acceptable as explained by the references include in the manuscript in lines 183-185, pages 9.

The considerations for the sample sizes were:

HIV positive women interviews: At each of the six health facilities, we planned to conduct in-depth interviews with 10 purposively selected women including two women in each of the following categories; women who declined ART, women having retention challenges, women having adherence challenges as well as women who were able to achieve very good retention and optimal adherence. Overall, 60 women were to be interviewed. However, 57 women were interviewed. Lines 164-168, page 9.

Health provider interviews: We planned that at each of the six facilities, we would conduct key informant interviews to two providers in each of the following categories; nurses/counselors, doctors, and expert clients. We also planned to interview the overall facility manager, and managers of the HIV, ANC/PMTCT, and immunization clinics, where these units were separate. This gave us a total of 60 interviews (10 at each facility). We were able to interview 54 health providers. Lines 178-180, page 9.

4. Were repeat interviews carried out? If yes, how many?

Dear Reviewer, we did not conduct repeat qualitative interviews.

5. Five of the lead investigators (A.M., R.W., E.B., R.N., and F.M.) interviewed the KIs: please spell KIs in full. Limit the use of abbreviations to the key terms in the study. KIIs: please spell in full. This is confusing to have KIs and KIIs. Just spell them in full.

Thank you for the above suggestion. We have written KIs, KIIs, IDI and IDIs in full in the entire manuscript

6. ‘’The specific questions to the key informants were ……’’ Please include interview schedule used in this study as appendix to the manuscript.

Thank you for the recommendation, we have included the key informant interview guide as S1 Appendix.

7. ‘’The questions to the IDI participants included;….’’ this level of information is not required in the manuscript. Please include interview schedule used in this study as appendix to the manuscript.

We have removed the detailed level of information from the manuscript (Lines 222-246 on pages 11-12) and included the in-depth interview guide as S2 Appendix.

Data collection tools and measures

1. ‘’……. written informed consent to the eligible women, and subsequently conducted face-to-face interviews using structured pre-tested and standardized questionnaires. Can the author briefly state how this was done?

The research assistants administered a written informed consent form to every eligible woman in a language which the woman best understood. Every woman who voluntarily accepted to participate in the study consented. Thereafter, the research assistants conducted the interviews in a quiet and private place using pre-tested and standardized questionnaires. Pre-tested and standardized questionnaires have been explained under question 2 above under “quantitative questionnaire” and included in the manuscript. Please see lines 142-146, page 8.

Results

1. Overall, in the three facilities where the quantitative study was conducted, how many women were eligible to participate in this study, how many declined to participate and how many eventually participated?

We thank the Reviewer for this question. Overall, 556 women were eligible for the study, 49 women declined to participate and 507 women were enrolled into the study as shown in figure 1. 

Discussion

1. ‘’HIV positive pregnant women should not be rushed to start on Option B+ ARVs on the same day of HIV diagnosis but be prepared well.’’ Can the authors elaborate on what they mean by ‘’ but be prepared well?’’

We have clarified on what we wanted to mean by “but be prepared well”. The women should be adequately counseled and supported. Please see lines 639-640 on page 40.

2. ‘’Facilities should have mechanisms to detect women who are nonadherent early enough so that they can be given supportive adherence counseling [24].’’ Authors should discuss a few of these mechanisms.

Dear Reviewer, we have discussed some of the proposed mechanisms as shown in lines 640-643 on page 40.

Strength and Limitations

1. Are there any other limitations, perhaps associated with the methodology (e.g. Individual interviews, etc)?

Dear Reviewer, we have added a limitation in regard to the methodology and explained how we minimized it. Lines 681-684, page 42.

2. A 30 days recall method was used to assess adherence. Recall bias was not mentioned as your limitation?

Thank you, Reviewer, for the addition. We have added recall bias as one of the limitations as explained in lines 679-681, page 42.

Conclusion and recommendation.

1. The authors should have a separate section for conclusion and recommendation. In addition, authors should discuss/provide the implications of their findings for practice, public health and policy.

Dear Reviewer, we have separated the section for conclusion and recommendations as shown on lines 685-700 pages 42-43. Implications of our study findings for practice, public health and policy have been provided. Please see lines 657-668 on page 41.

Others

1. Authors should discuss the generalisability (external validity) of the study results.

Our study findings can be generalizable to similar settings like regional referral hospitals, general hospitals and HC IIIs. We have included this is the discussion of the study results. Lines 649-656, pages 40-41. 

Reviewer #2: Well written manuscript with adequate background information about the problem, study gap or rationale, and study objectives. Appropriate methods as regards study design, selection criteria, setting, data sources, measurements, operational definitions, human subject and ethical considerations, and analysis were articulated.

However, few details may need to be addressed to improve clarity and interpretation findings of the manuscript:

Methods:

1. The authors referenced parent study for sample size. However, it is critical for the authors to have demonstrated whether the parent sample size (references 28 and 29) was adequate in precision or power to detect the desired outcomes for the findings in this manuscript – uptake, early adherence and associated factors. Assumptions and power calculations could have been stated.

Thank you so much Reviewer.

The sample size for the parent study was 507 and was determined as described in our earlier paper by Naigino, R., Makumbi, F., Mukose, A., Buregyeya, E., Arinaitwe, J., Musinguzi, J. and Wanyenze, R.K., 2017. HIV status disclosure and associated outcomes among pregnant women enrolled in antiretroviral therapy in Uganda: a mixed methods study. Reproductive Health, 14(1), p.107. However, for this manuscript we used a sample size of 463 was used for the Uptake whereas 410 was used for early adherence and associated factors. This is the number of women who had complete data. 

A 5% precision was determined using the sample sizes for the two study outcomes (uptake and early adherence). We have included this in the manuscript in lines 156-158 on page 8.

In terms of power calculation, adherence was at 50% among the 56 participants who were not ready to start ART immediately and 83.23% among the 310 who were ready to start immediately. From power calculation this gives 99.99% power to detect a difference in adherence between the two groups. Adherence was at 84.14 % among the 290 participants who had disclosed their HIV status to anyone and 55.26% among the 76 who had not disclosed. From power calculation for this objective, it gives 99.88% power to detect a difference in adherence between the two groups. Therefore, the study had sufficient power (>80%). We have included this in the manuscript in lines 158-160, page 8.

2. In the analytic strategy, it was important for the authors to have stated whether they explored for interaction or effect modification for certain factors on to the outcome, particularly facility level interacting with other factors. Participants and factors at health center IV could have been different at general and referral hospitals.

Dear Reviewer, thank you so much for these very critical comments. We had considered the issues raised but we hadn’t included it in the manuscript. We have now included it.

We found that there was no interaction for factors explored. These included: disclosure of HIV status to anyone; readiness to start ART; health facility level and knowledge on how long ART should be taken. Lines 261-265 on page 13. 

Results: The results are relevant to the problem posed. Synthesis of results was satisfactory. However, the presentation may need to be improved.

1. A table showing how comparable the select baseline variables in Table 1 from participants who declined (49 in figure 1) versus those who were included in the study. This would support interpretations on external validity of the findings.

Thank you so much Reviewer, 49 eligible women refused to participate in the study. Unfortunately, we don’t have baseline characteristics for 8 women who didn’t provide these characteristics on the screening form. The characteristics for the 41 women have been included in the manuscript in lines 313-318, page 17-18 and Table 2 in lines 320-322 on pages 18-19. Comparison of the characteristics of the women who were enrolled in the study with those who declined has been included in lines 323-325 on page 19 and in table 3 in lines 326-328 on pages 19-21. The comparison shows that the participants who were enrolled and those who declined did not vary by baseline characteristics except for the categories of casual worker and housewife under the variable “Occupation”. 

2. Table 6 shows results with appropriate analytic strategy; however, certain baseline variables in Table 1 were not considered. Further, and as already discussed by the authors that uptake and adherence may differ by health facility level, authors could have explored possible interactions between health facility level versus certain baseline variables and variables in Table 6 onto the outcome. Findings from exploring interactions would support the need for certain targeted interventions at different levels of the health facility as compared to the general recommendations that have been articulated in the discussion.

Thank you, Reviewer, for this critical question.

Yes, some of the baseline characteristics don’t appear in table 6. They were dopped during model building using a criterion that included; testing for collinearity, interaction, confounding and the P-value of ≤ 0.2 at bivariate analysis. This has been included in the manuscript in lines 261-265, page 13.

Interpretation and conclusions: The discussion was balanced with articulation of the limitations and strengths. The explanations or theories were appropriate to the claims.

1. However, the authors may need to revise the discussion when results change in line with the suggestions I have made in the methods and result sections.

Thank you so much Reviewer, as explained above, we explored for interaction and there was none. This has been incorporated in the manuscript as shown in lines 261-265, page 13.

2. The limits of the external validity or generalizability were not explicitly discussed

Our study findings can be generalizable to similar settings like regional referral hospitals, general hospitals and HC IIIs. We have included this is the discussion of the study results. Please see lines 649-656, pages 40-41.

 6. PLOS authors have the option to publish the peer review history of their article (what does this mean?). If published, this will include your full peer review and any attached files.

Do you want your identity to be public for this peer review? For information about this choice, including consent withdrawal, please see our Privacy Policy.

Reviewer #1: No

Reviewer #2: Yes: Ezekiel Mupere MBChB, MMed, MS., PhD

---

## [Decision Letter · Decision Letter 1]

31 Mar 2021

PONE-D-20-22840R1

What influences uptake and early adherence to Option B+ (lifelong antiretroviral therapy among HIV positive pregnant and breastfeeding women) in Central Uganda? A mixed methods study

PLOS ONE

Dear Dr. Mukose,

Thank you for submitting your manuscript to PLOS ONE. After careful consideration, we feel that it has merit but does not fully meet PLOS ONE’s publication criteria as it currently stands. Therefore, we invite you to submit a revised version of the manuscript that addresses the points raised during the review process.

I have gone through the manuscript for final edits. Please review the version of the manuscript in which I have made changes and comments. Please add explanations as needed and send back your final edited version. 

We look forward to receiving your revised manuscript.

Kind regards,

Julie AE Nelson, PhD

Academic Editor

PLOS ONE

Journal Requirements:

Reviewers' comments:

Reviewer's Responses to Questions

**Comments to the Author**

1. If the authors have adequately addressed your comments raised in a previous round of review and you feel that this manuscript is now acceptable for publication, you may indicate that here to bypass the “Comments to the Author” section, enter your conflict of interest statement in the “Confidential to Editor” section, and submit your "Accept" recommendation.

Reviewer #1: All comments have been addressed

Reviewer #2: All comments have been addressed

2. Is the manuscript technically sound, and do the data support the conclusions?

Reviewer #1: Yes

Reviewer #2: Yes

3. Has the statistical analysis been performed appropriately and rigorously? 

Reviewer #1: I Don't Know

Reviewer #2: Yes

4. Have the authors made all data underlying the findings in their manuscript fully available?

Reviewer #1: Yes

Reviewer #2: Yes

5. Is the manuscript presented in an intelligible fashion and written in standard English?

Reviewer #1: Yes

Reviewer #2: Yes

6. Review Comments to the Author

Reviewer #1: The authors should place the recommendation section in the manuscript before conclusion. Thank you. I have no further comments.

Reviewer #2: Thank you very much for taking time to address the review comments. The revised manuscript and revision comments are well received.

7. PLOS authors have the option to publish the peer review history of their article (what does this mean?). If published, this will include your full peer review and any attached files.

Reviewer #1: **Yes: **Dr Olumuyiwa Omonaiye

Reviewer #2: **Yes: **Ezekiel Mupere MBChB, MMed, MS., PhD (Epidemiology & Bio-statistics)

---

## [Author Response · Author response to Decision Letter 1]

6 Apr 2021

PONE-D-20-22840R1

What influences uptake and early adherence to Option B+ (lifelong antiretroviral therapy among HIV positive pregnant and breastfeeding women) in Central Uganda? A mixed methods 

PLOS ONE

Dear Dr. Mukose,

Thank you for submitting your manuscript to PLOS ONE. After careful consideration, we feel that it has merit but does not fully meet PLOS ONE’s publication criteria as it currently stands. Therefore, we invite you to submit a revised version of the manuscript that addresses the points raised during the review process.

I have gone through the manuscript for final edits. Please review the version of the manuscript in which I have made changes and comments. Please add explanations as needed and send back your final edited version. 

Response to comments: 

Dear Reviewers and the Academic Editor, we thank you for your time to review our responses. We also thank you the Academic Editor for going through the manuscript for final edits.

We have added the explanations as needed. Please find detailed point by point explanations and the changes highlighted in the “Revised Manuscript with Track Changes”.

Dear the Academic Editor, please find detailed point by point explanations. 

1. Not clear what these mean (HC IV, HC III). Are these designations necessary? Add explanation if needed, delete if not needed. Be consistent throughout and define all or delete all 

Dear Editor, according to the structure of the Uganda public health-care system, there are seven levels of health care delivery, organized from lower to higher levels in a hierarchy. HC IV means health center 4 whereas HC III refers to health center 3. A reference (Mukose, Aggrey David, et al. "Health Provider Perspectives of Health Facility Preparedness and Organization in Implementation of Option B+ among Pregnant and Lactating Women in Central Uganda: A Qualitative Study." Journal of the International Association of Providers of AIDS Care (JIAPAC) 18 (2019): 2325958219833930) has been included. Please see lines 124-125 on page 7

2. Explain the modifications. Not clear whether the references indicate the modified or unmodified questionnaire. 

The references indicate the unmodified questionnaire. We picked questions and issues that were used in the references. They were used to come up with the final questionnaire that was used our study. The modifications ensured that the questionnaire was context specific.

3. A 5% precision was determined using the sample sizes that we used for the two study outcomes (levels of uptake, and early adherence). Unclear what this means. Please clarify

Dear Academic Editor, we have clarified the sentence to read as: Basing the sample sizes that we used for the two study outcomes, (levels of uptake; and early adherence), the study had a 5% level of precision. Lines 157-159, page 8.

4. Analysis revealed that there was no interaction.

Thank you so much Editor we have moved the sentence “Analysis revealed that there was no interaction” from the methods section (Line 240, page 12) to the results section; (line 510, page 32).

5. Relative risks are indicated, but no interactions were found among factors discussed below. This needs to be in the results somewhere. This is just a suggestion. Could add a small section at the end of the results instead (Page 13).

Thank you so much. This has been modified and added to the section on “Factors associated with optimal early adherence". Please see lines 509 -510 on page 32.

6. 05_IDI, 02_IDI etc.

“05_IDI” was used to link the quote to the in-depth interview participant. We have replaced it with “IDI Participant” throughout the manuscript.

7. Academic Editor: Maybe move this paragraph (Recommendations) to the end of the Implications section and delete the header.

 Reviewer #1: The authors should place the recommendation section in the manuscript before 

 conclusion.

Thank you so much the Academic Editor and Reviewer #1. We have moved the section on recommendations from a separate section (Lines 674 -680 on page 42) to the end of the implications section and deleted the header. Please see lines 643-648, page 40.

We look forward to receiving your revised manuscript.

Kind regards,

Julie AE Nelson, PhD

Academic Editor

PLOS ONE 

Journal Requirements:

.

Reviewers' comments:

Reviewer's Responses to Questions

Comments to the Author

1. If the authors have adequately addressed your comments raised in a previous round of review and you feel that this manuscript is now acceptable for publication, you may indicate that here to bypass the “Comments to the Author” section, enter your conflict of interest statement in the “Confidential to Editor” section, and submit your "Accept" recommendation.

Reviewer #1: All comments have been addressed

Reviewer #2: All comments have been addressed

2. Is the manuscript technically sound, and do the data support the conclusions?

Reviewer #1: Yes

Reviewer #2: Yes

3. Has the statistical analysis been performed appropriately and rigorously?

Reviewer #1: I Don't Know

Reviewer #2: Yes

4. Have the authors made all data underlying the findings in their manuscript fully available?

Reviewer #1: Yes

Reviewer #2: Yes

5. Is the manuscript presented in an intelligible fashion and written in standard English?

Reviewer #1: Yes

Reviewer #2: Yes

6. Review Comments to the Author

Reviewer #1: The authors should place the recommendation section in the manuscript before conclusion. Thank you. I have no further comments.

Thank you so much the Academic Editor and Reviewer #1. We have moved the section on recommendations from a separate section (Lines 674 -680 on page 42) to the end of the implications section and deleted the header. Please see lines 643-648, page 40.

Reviewer #2: Thank you very much for taking time to address the review comments. The revised manuscript and revision comments are well received.

7. PLOS authors have the option to publish the peer review history of their article (what does this mean?). If published, this will include your full peer review and any attached files.

Do you want your identity to be public for this peer review? For information about this choice, including consent withdrawal, please see our Privacy Policy.

Reviewer #1: Yes: Dr Olumuyiwa Omonaiye

Reviewer #2: Yes: Ezekiel Mupere MBChB, MMed, MS., PhD (Epidemiology & Bio-statistics)

Thank you so much Reviewers for accepting all our responses.

---

## [Editor Report · Decision Letter 2]

22 Apr 2021

What influences uptake and early adherence to Option B+ (lifelong antiretroviral therapy among HIV positive pregnant and breastfeeding women) in Central Uganda? A mixed methods study

PONE-D-20-22840R2

Dear Dr. Mukose,

We’re pleased to inform you that your manuscript has been judged scientifically suitable for publication and will be formally accepted for publication once it meets all outstanding technical requirements.

Kind regards,

Julie AE Nelson, PhD

Academic Editor

PLOS ONE
---

## [Editor Report · Acceptance letter]

26 Apr 2021

PONE-D-20-22840R2 

What influences uptake and early adherence to Option B+ (lifelong antiretroviral therapy among HIV positive pregnant and breastfeeding women) in Central Uganda? A mixed methods study 

Dear Dr. Mukose:

I'm pleased to inform you that your manuscript has been deemed suitable for publication in PLOS ONE. Congratulations! Your manuscript is now with our production department. 

Kind regards, 

on behalf of

Dr. Julie AE Nelson 

Academic Editor

PLOS ONE